# Anti-leishmanial physalins—Phytochemical investigation, *in vitro* evaluation against clinical and MIL-resistant *L. tropica* strains and *in silico* studies

**Saira Bano[1], Memoona Bibi[1], Saba Farooq[2], Humaira Zafar[2], Muniza Shaikh[2], Behram Khan Khoso[3], Sammer Yousuf [1]\*, M. Iqbal Choudhary[1,2,4]\***

**1** H.E.J. Research Institute of Chemistry, International Center for Chemical and Biological Sciences, University of Karachi, Karachi, Pakistan, **2** Dr. Panjwani Center for Molecular Medicine and Drug Research, International Center for Chemical and Biological Sciences, University of Karachi, Karachi, Pakistan, **3** Jinnah Post-Graduate Medical Center, Karachi, Pakistan, **4** Department of Biochemistry, Faculty of Science, King Abdulaziz University, Jeddah, Saudi Arabia

\* dr.sammer.yousuf@gmail.com (SY); iqbal.choudhary@iccs.edu (MIC)

**Data Availability Statement:** All relevant data are within the paper and its Supporting Information files.

## Abstract

Cutaneous leishmaniasis (CL) is a major health problem in over 98 countries of the world, including Pakistan. The current treatments are associated with a number of adverse effects and availability problem of drugs. Therefore, there is an urgent need of easily available and cost effective treatments of CL- in Pakistan. The bioassay-guided fractionation and purification of crude extract of *Physalis minima* has led to the isolation of a new aminophysalin B (**1**), and eight known physalins, physalin B (**2**), 5ß,6ß-epoxyphysalin B (**3**), 5α-ethoxy-6ß-hydroxy-5,6-dihydrophysalin B (**4**), physalin H (**5**), 5ß,6ß-epoxyphysalin C (**6**), and physalin G (**7**), K (**8**), and D (**9**). It is worth noting that compound **1** is the second member of amino-physalin series, whereas compound **6** was fully characterized for the first time. The structures of compounds **1–9** were elucidated by spectroscopic techniques Whereas, the structural assignments of compounds **1** and **8** were also supported by single-crystal X-ray diffraction studies. The anti-leishmanial activity of isolated physlains **1–9** was evaluated against *Leishmania major* and *Leishmania tropica* promastigotes. Compounds **2**, **3**, and **5–7** ($IC_{50}$ = 9.59 ± 0.27–23.76 ± 1.10 µM) showed several-fold more potent activity against *L. tropca* than tested drug miltefosine ($IC_{50}$ = 42.75 ± 1.03 µm) and pentamidine ($IC_{50}$ = 27.20 ± 0.01 µM). Whereas compounds **2**, **3** and **5** ($IC_{50}$ = 3.04 ± 1.12–3.76 ± 0.85 µM) were found to be potent anti-leishmanial agents against *L. major*, several fold more active than tested standard miltefosine ($IC_{50}$ = 25.55 ± 1.03 µM) and pentamidine ($IC_{50}$ = 27.20 ± 0.015 µM**)**. Compounds **4** ($IC_{50}$ = 74.65 ± 0.81 µM) and **7** ($IC_{50}$ = 39.44 ± 0.65 µM) also showed potent anti-leishmanial ativity against the miltefosine-unresponsive *L. tropica* strain (MIL resistant) (miltefosine $IC_{50}$ = 169.55 ± 0.78 µM). Molecular docking and predictive binding studies indicated that these inhibitors may act *via* targeting important enzymes of various metabolic pathways of the parasites.

**Funding:** "Sammer Yousuf and M. Iqbal Choudhary, researcher leading to these results has, in part, received funding from UK Research and Innovation via the Global Challenges Research Fund under grant agreement 'A Global Network for Neglected Tropical Diseases', grant number MR/P027989/1." "The funders had no role in study design, data collection and analysis, decision to publish, or preparation of the manuscript."

**Competing interests:** The authors have declared that no competing interests exist.

# Introduction

Cutaneous leishmaniasis, a neglected tropical disease (NTD), infects 700,000 to 1.2 million poorest people in over 98 countries of Africa, Asia, the Middle East and America annually. Poverty, migration, poor hygiene, and malnutrition are the major risk factors associated with endemicity of various forms of leishmaniasis in the developing world [1–3]. About 70 types of sand flies are known to be responsible to transmit the *Leishmania* parasite geographically distributed Old (Asia, Africa, and the Middle East) and New world *i.e.* western hemisphere countries. Broadly they belong to the phlebotomine and Lutzomyia genera. The diagnosis of leishmaniasis, both parasitic (direct) and immunological (indirect), is often challenging in low-resource countries [1]. Pakistan is among the endemic countries with CL cases in almost every part of the country. In 2020, as per WHO Global Leishmaniasis Surveillance Report, Afghanistan, Algeria, Brazil, Colombia, Iraq, Pakistan and Syria, are the seven countries, each reported >6,000 CL cases, representing >80% of cases globally [4]. CL is not a life-threatening infection but cause extreme psychological trauma, and stigma as infected people pushed to be excluded from public life.

The available therapeutic options are now losing effectiveness due to the emergence of resistance [5, 6]. Furthermore, these available drugs are expensive, and some have toxic effects on liver and kidneys [7]. Medicinal plants have immense potential to yield compounds that are capable of treating many ailments. In recent days, pharmaceutical industries have reformed their attention towards plant-based medicine and formulation because of their least toxicity and more efficacy [8–10]. *Physalis minima* Linn is a member of the *Solanaceae* family, commonly known as native gooseberry, and reported for anti-cancer, anti-diabetic, anti-oxidant, anti-ulcer, analgesic, antipyretic, inhibition of amylase, lipase and alpha-glucosidase enzymes, as well as anti-gonorrhoeal, and anti-inflammatory properties [11–15]. Previous research on *P. minima* led to the isolation of bioactive withanolides and physalins, which showed potent anti-inflammatory and cytotoxic activities [16–19]. We have reported the *in vitro* anti-leishmanial activity of the *Physalis minima* plant extracts and its pure constituents physalin B (**2**), 5ß,6ß-epoxyphysalin B (**3**), and physalin H (**5**) against promastigotes of *L. major* for the first time [20–22]. The presented study deals with the isolation, structure determination, and anti-leishmanial effect of a new aminophysalin B (**1**) along with eight known physalins, physalin B (**2**), 5ß,6ß-epoxyphysalin B (**3**), 5α-ethoxy-6ß-hydroxy-5,6-dihydrophysalin B (**4**), physalin H (**5**), 5ß,6ß-epoxyphysalin C (**6**), and physalin G (**7**), K (**8**), and D (**9**) *in vitro* against promastegotes of *L. tropica*. Compounds **1, 4, 6, 7, 8** and **9** were also evaluated against promastegotes of *L. major* but no significant activity was observed. Moreover, the effect of compounds **2–9** were also evaluated for their potential against the MIL- resistant *L. tropica*. These steroidal lactones **1–9** were also subjected to in-silico studies to predict their mechanisms of action.

# Materials and methods

## General

Precoated TLC plates (silica gel, 20×20, 0.25 mm thick PF$_{254}$, Merck, Germany) were used for thin layer chromatography, and stained by spraying with Dragendroff's and vanillin reagents. Column chromatography was carried out on silica gel (70–230 mesh, Merck, Germany) [23]. Recycling preparative separation was performed on a HPLC JAI LC-908W (Japan) instrument, equipped with GAIGEL-SIL, D-60-10, CHCl$_3$-MeOH as the mobile phase, with UV detection at 254 nm. JMS-600H mass spectrometer (JEOL, Japan) was used to record the mass spectra (EI- and HREI-MS) in *m*/z (relative abundance %). Electrospray ionization mass spectra (ESI-MS) were recorded on Amazon speed ion trap low-resolution mass spectrometer (Bruker

Daltonics, Germany) and high resolution electrospray ionization mass spectra (HRESI-MS) were measured on MaXis II ESI–QTOF Ultra High-Resolution Mass Spectrometer (Bruker Daltonics, Germany).Ultraviolet (UV) experiments were performed in methanol on Hitachi U-3200 spectrophotometer (Hitachi, Japan). Infrared (IR) analysis was performed in KBr on FT-IR-8900 (Shimadzu, Japan), and Bruker VECTOR 22 spectrophotometers (Bruker, France). UV Spectra (nm) were measured in methanol and chloroform on Evolution 300 UV visible spectrophotometer. FT-IR-8900 spectrophotometer was used to record IR spectra (cm$^{-1}$) (JASCO, Japan). Optical rotations were measured in methanol with a digital polarimeter JASCO P-2000 (JASCO, Japan) by using 10 cm cell tube. $^1$H- and $^{13}$C-NMR spectra were recorded on Avance NMR spectrometer at 500 and 75–150 MHz in CDCl$_3$ or CD$_3$OD, respectively (Bruker, Zurich, Switzerland). Standard pulse sequences were used for DEPT, and 2D-NMR experiments. The $^1$H-NMR, $^{13}$C-NMR, COSY, HMBC, HMQC, and NOESY spectra were recorded in deuterated chloroform, and methanol on Bruker Avance spectrometers (Bruker, Zurich, Switzerland). X-Ray crystallography data were collected on Bruker D8 venture diffractometer. Cu Kα radiation (1.54178 Å) was used for X-ray diffraction data collection. SAINT program was employed for data integration and reduction; Direct method was utilized for structure solution with full-matrix least square method with aid of SHELXTL (SHELX2016/6).

## Plant material

Fresh plants (50 kg.) of *Physalin minima* (L. Var. indica), were collected from Karachi, Pakistan, and identified by a plant taxonomists, Department of Botany, University of Karachi. A voucher specimen (G. H. 68261) was deposited at the herbarium.

## Extraction and isolation

The air-dried plants (11 kg) were extracted with EtOH (100 L) for 10 days. After evaporation of the solvent, a brownish residue (1.7 kg) was obtained that was allowed to be suspended in distilled H$_2$O (5 L), and defatted with petroleum ether (10 L) [24]. The defatted aqueous extract was further extracted with CH$_2$Cl$_2$ (35 L). The CH$_2$Cl$_2$ extract was concentrated under reduced pressure to obtain a greenish dried powder (85 g) that was subjected to column chromatography and eluted with Pet ether: CH$_2$Cl$_2$, CH$_2$Cl$_2$, CH$_2$Cl$_2$: MeOH (10:90. . .90:10, 100, 99:1–97:3) to afford four main fractions **A** (2.14 **g**), **B** (1.54 g), **C** (1.14 g), and **D** (24 g), respectively. The repeated column chromatography of fraction **A** (2.14 g) afforded one major fraction **a** (300 mg, petroleum: acetone 80: 20) which was further purified on recycling reverse phase HPLC to obtain purified compound **2** (10 mg, CHCl$_3$: MeOH-99:1, Flow rate: 4 mL/min, R$_T$ = 16 min) and **3** (60 mg, CHCl$_3$: MeOH-99:1, Flow rate: 4 mL /min, R$_T$ = 18min). Fraction **B** (1.54 **g**) yielded sub fraction **b** (500 mg, petroleum: acetone 70: 30) which was further subjected to recycling reverse phase HPLC to obtain compound **5** (10 mg, CHCl$_3$: MeOH-99:1, Flow rate: 4 mL/min, R$_T$ = 16 min). Fraction **C** (1.14 **g**) afforded one major sub fraction **c** (petroleum: acetone 80: 20) which yielded and purified compound **6** (5 mg, CHCl$_3$: MeOH-99:1, Flow rate: 4 mL/min, R$_T$ = 16 min) *via* recycling reverse phase HPLC. Fraction **D** (24 **g**) was subjected to column chromatography and three major sub fractions were obtained. Subtraction **d** (1.41) (petroleum ether: acetone-55:45), **e** (1.0) (Petroleum ether: acetone-45:55) and **f** (20 g) (petroleum: acetone-20:80) which were purified by using recycling reverse phase HPLC to obtain compounds **1** (3 mg, CHCl$_3$: MeOH-97:3, Flow rate: 3 mL/min, R$_T$ = 24 min) and **4** (4 mg, CHCl$_3$: MeOH-96:4, Flow rate: 4 mL/min, R$_T$ = 24 min), **7** (10mg, CHCl$_3$: MeOH-97:3, Flow rate: 3 mL/min, R$_T$ = 30 min), **8** (6 mg, CHCl$_3$: MeOH-97:3, Flow rate: 3 mL/min, R$_T$ = 24 min), and **9** (6 mg, CHCl$_3$: MeOH-97:3, Flow rate: 3 mL/min, R$_T$ = 32 min.

**Aminophysalin B (1).** Colorless solid (3 mg): $[\alpha]_D$ = -65 (c = 0 .01, MeOH), **UV** (MeOH) $\lambda_{max}$ nm (log∈): 312 (3.2). **IR** (KBr) $\nu_{max}$ cm$^{-1}$: 3463 (OH), 1732 (C = O), 1656 (C = C-C = O), **1H-NMR** (CDCl$_3$, 500 MHz): Table 1. **13C-NMR** (CDCl$_3$, 175 MHz): Table 1. **HRESI-MS** *m/z*: 526.2059 [M+H]$^+$(C$_{24}$H$_{31}$NO$_9$+H requires 526.2072).

*Single-crystal X-ray diffraction data.* Empirical formula = C$_{28}$ H$_{31}$ NO$_9$, H$_2$O, Mr = 543.55, Crystal system: orthorhombic, space group: P2$_1$2$_1$2$_1$, Unit cell dimensions: **a** = 9.7870(4) Å, **b** = 9.7870(4) Å, **c** = 16.3581(7) Å, Volume: 2412.66(18) Å$^3$, Z = 4, $\rho_{calc}$ = 1.496 Mg/m3, F(000) 1152, Crystal size: 0.900 x 0.060 x 0.030 mm, θ range for data collection: 5.266 to 68.135 deg. Total 15,452 reflections were collected, out of which 4,364 were found unique (R$_{int}$ = 0.0945). Final R indices were R$_1$ = 0.0551for [I>2σ (I)], wR$_2$ = 0.1446, R indices were R$_1$ = 0.0699, wR$_2$ = 0.1547 for all data, largest diff. peak and hole: 0.434 and -0.369 e. Å$^{-3}$, CCDC No. 2177422.

**Physalin B (2).** Colorless solid (10 mg): $[\alpha]_D$ = -257 (c = 0 .05, MeOH), **UV** (MeOH) $\lambda_{max}$ nm (log∈): 230 (3.6). **IR** (KBr) $\nu_{max}$ cm$^{-1}$: 3410 (OH), 1748 (lactone), 1653 (C = C-C = O), **1H-NMR** (CDCl$_3$, 500 MHz): 1.19 (s, H$_3$-28), 1.25 (s, H$_3$-19), 1.95 (s, H$_3$-21), 4.35 (t, overlapped $J_{22,23}$ = 2.5 Hz, H-22), 3.75 (d, $J_{27,27}$ = 12.5 Hz, H$_2$-27), 4.50 (dd overlapped, $J_{27,27}$ = 13.5 Hz, $J_{27,25}$ = 4.5, H$_2$-27), 5.91 (dd, $J_{2,3}$ = 10.0 Hz, $J_{2,4ß}$ = 2 Hz, H-2), 6.77 (ddd, $J_{3,2}$ = 10.0

**Table 1. 1H-NMR and 13C-NMR chemical shift data of compounds 1 and 6 (δ in ppm, *J* in Hz).**

| C. No. | 1$^a$ | 1$^b$ | 6$^a$ | 6$^b$ |
|---|---|---|---|---|
| 1 | - | 208.8 | - | 204.4 |
| 2 | 6.00 (d, $J_{2,3}$ = 10.0) | 119.8 | 5.99 (dd, $J_{2,3}$ = 10.0, $J_{2,4}$ = 2.5) | 145.5 |
| 3 | 6.98 (dd, $J_{3,4}$ = 9.5, $J_{3,4}$ = 6.0) | 140.9 | 6.83 (ddd, $J_{3,2}$ = 10.0, $J_{3,4ß}$ = 2.5, $J_{3,4\alpha}$ = 6.0) | 128.6 |
| 4 | 6.12 (d, $J_{3,4}$ = 5.5) | 125.5 | 2.93, m; 2.04, m | 32.9 |
| 5 | - | 155.6 | - | 61.8 |
| 6 | 4.49, m | 70.4 | 3.24 (d, $J_{3,4}$ = 2.5) | 64.6 |
| 7 | 2.10, m; 2.53, m | 29.4 | 2.35, m; 1.71, m | 29.8 |
| 8 | 3.13, m | 39.4 | 2.22, m | 39.1 |
| 9 | 2.57, m | 34.3 | 2.51, m | 35.5 |
| 10 | - | 52.0 | - | 49.8 |
| 11 | 2.35, m; 1.65, m | 28.2 | 2.26, m; 1.14, m | 22.4 |
| 12 | 2.54, m; 2.04, m | 29.5 | 2.64, m; 1.76, m | 26.4 |
| 13 | - | 79.4 | - | 79.7 |
| 14 | - | 210.7 | - | 102.1 |
| 15 | - | 171.5 | - | 210.8 |
| 16 | 2.52, m | 56.5 | 2.50, m | 54.6 |
| 17 | - | 83.3 | - | 81.8 |
| 18 | - | 173.0 | - | 171.8 |
| 19 | 1.54, s | 23.0 | 1.21, s | 14.3 |
| 20 | - | 80.0 | - | 82.7 |
| 21 | 1.85, s | 21.7 | 1.92, s | 20.9 |
| 22 | 4.50, m | 77.3 | 4.58, (dd, $J_{22,23}$ = 4.5, $J_{22,23}$ = 2) | 76.8 |
| 23 | 2.05, m; 1.97, m | 31.6 | 1.99, m; 2.11, m | 31.7 |
| 24 | - | 29.1 | - | 36.6 |
| 25 | 2.36, m | 42.2 | - | 139.3 |
| 26 | - | 164.9 | - | 163.7 |
| 27 | 4.17 (dd, $J_{27,27}$ = 10.0, $J_{27,25}$ = 20.5) 4.46 ($J_{27,27}$ = 21) | 47.7 | 4.20 (6.64, s; 5.72, s) | 133.3 |
| 28 | 1.04, s | 28.2 | 1.58, s | 27.3 |

$^a$ 500 MHz, CDCl$_3$

Hz, $J_{3,4a}$ = 5.0 Hz, $J_{3,4ß}$ = 2.5, H-3), 5.56 (d, $J_{6,7}$ = 6.0 Hz, H-6), EIMS *m/z* (rel.int. %): 510 (M$^+$), 492 (100.0), 477 (21.7), 464 (48.0), 173 (46.2), 159 (70.9), 145 (26.1), 91 (26.6), 43 (18.1).

**5β, 6β-epoxyphysalin B (3).** White solid (60 mg), [α] $_D$ = -108 (*c* = 0 .05, MeOH), **UV** (MeOH) λ$_{max}$ nm (log∈): 230 (3.5). **IR** (KBr) ν$_{max}$ cm$^{-1}$: 3422 (OH), 1778 (lactone), 1656 (C = C-C = O), **$^1$H-NMR** (CDCl$_3$, 500 MHz): 1.24 (s, H$_3$-28), 1.29 (s, H$_3$-19), 1.93 (s, H$_3$-21), 3.25 (d, $J_{6α, 7β}$ = 3.0 Hz, H-6), 4.52 (t, overlapped $J_{22,23}$ = 1.5 Hz, H-22), 3.74 (d, $J_{27,27}$ = 13.5 Hz, H-27), 4.50 (dd overlapped, $J_{27,27}$ = 14.0 Hz, $J_{27,25}$ = 4.5 Hz, H-27), 5.99 (dd, $J_{2,3}$ = 10.5 Hz, $J_{2, 4ß}$ = 2.0 Hz, H-2), 6.84 (ddd, $J_{3,2}$ = 10.0 Hz, $J_{3,4α}$ = 5.0 Hz, $J_{3,4ß}$ = 2.5 Hz, H-3), EI-MS *m/z* (rel. int. %): 526 (6.7, M$^+$), 498 (100.0), 480 (15.7), 360 (16.6), 159 (17. 0), 133 (20.6), 91 (14.6), 55 (15.0).

**5α-ethoxy-6ß-hydroxy-5,6-dihydrophysalin B (4).** Colorless solid (4 mg): [α] $_D$ = -58 (*c* = 0 .05, MeOH), **UV** (MeOH) λ$_{max}$ nm (log∈): 230 (3.2). **IR** (KBr) ν$_{max}$ cm$^{-1}$: 3405 (OH), 1787 (lactone), 1667 (C = C-C = O), **$^1$H-NMR** (CDCl$_3$, 500 MHz): 0.98 (t, *J* = 7 Hz, $\underline{CH_3}CH_2O$), 1.24 (s, H$_3$-28), 1.50 (s, H$_3$-19), 1.98 (s, H$_3$-21), 4.00 (m, H-6), 4.54 (t, overlapped $J_{22,23}$ = 1.5, H-22), 3.76 (d, $J_{27,27}$ = 13.5 Hz, H-27), 4.49 (dd, $J_{27,25}$ = 4.5 Hz H-27), 5.85 (dd, $J_{2,3}$ = 10.0 Hz, $J_{2, 4ß}$ = 2.5 Hz, H-2), 6.57 (ddd, $J_{3,2}$ = 10.0 Hz, $J_{3,4α}$ = 5.0 Hz, $J_{3,4ß}$ = 2.0 Hz, H-3), EI-MS *m/z* (rel. int. %): 572 (7.4, M$^+$), 544 (10.8), 526 (24.8), 508 (60.1), 454 (30.3), 171 (79.3), 147 (77.1), 133 (100.0), 109 (89.9), 91 (64.3), 55 (75.6).

**Physalin H (5).** Colorless solid (10 mg): [α] $_D$ = -260 (*c* = 0.01, MeOH), **UV** (MeOH) λ$_{max}$ nm (log∈): 230 (3.8). **IR** (KBr) ν$_{max}$ cm$^{-1}$: 3426 (OH), 1778 (lactone), 1673 (C = C-C = O), **$^1$H-NMR** (CDCl$_3$, 500 MHz):1.35 (s, H$_3$-19), 1.25 (s, H$_3$-28), 1.99 (s, H$_3$-21), 4.10 (d, $J_{6α, 7β}$ = 3.0 Hz, H-6), 4.54 (t, overlapped $J_{22,23}$ = 2.5 Hz, H-22), 3.75 (d, $J_{27,27}$ = 12.5 Hz, H$_2$-27), 4.50 (dd, overlapped, $J_{27,27}$ = 13.5 Hz, $J_{27,25}$ = 4.5 Hz, H$_2$-27), 5.94 (dd, $J_{2,3}$ = 10.0 Hz, $J_{2, 4ß}$ = 2.0 Hz, H-2), 6.71 (ddd, $J_{3,2}$ = 10.0 Hz, $J_{3,4α}$ = 5.0 Hz, $J_{3,4ß}$ = 2.5 Hz, H-4) **HRE-SI-MS** m/z: 563.1674 [M+H]$^+$ (C$_{28}$H$_{31}$ClO$_{10}$+H requires 563.1679).

**5β, 6β-epoxyphysalin C (6).** White solid (5 mg): [α] $_D$ = -168 (c = 0 .01, MeOH), **UV** (MeOH) λ$_{max}$ nm (log∈): 230 (3.1). **IR** (KBr) ν$_{max}$ cm$^{-1}$: 3427 (OH), 1782 (lactone), 1656 (C = C-C = O), **$^1$H-NMR** (CH$_3$OD, 500 MHz): <u>Table 1</u>. **$^{13}$C-NMR** (CH$_3$OD, 175 MHz): <u>Table 1</u>. **HRESI-MS** *m/z*: 527.1907 [C$_{29}$H$_{30}$O$_{10}$+H; 527.1912].

**Physalin G (7).** White solid (10 mg): [α] $_D$ = -92 (*c* = 0 .01, MeOH), **UV** (MeOH) λ$_{max}$ nm (log∈): 312 (5.8).**IR** (KBr) ν$_{max}$ cm$^{-1}$: 3418 (OH), 1776 (lactone), 1622 (C = C-C = O), **$^1$H-NMR** (CDCl$_3$, 500 MHz):1.25 (s, H$_3$-28), 1.93 (s, H$_3$-19), 1.51 (s, H$_3$-21), 4.49 (m, H-6), 4.52 (m, H-22), 3.74 ($J_{27,27}$ = 13.0 Hz, H$_2$-27), 4.50 (m, H$_2$-27), 5.93 (d, $J_{2,3}$ = 5.5 Hz, H-2), 6.82 (dd, $J_{3,2}$ = 9.5 Hz, $J_{3,4}$ = 5.5 Hz, H-4), 6.03 (d, $J_{4,3}$ = 6.0 Hz, H-4), **EIMS** *m/z* (rel. int. %): 526 (2.8, M$^+$), 508 (55.1), 482 (79.8), 454 (40.1), 185 (38.9), 159 (27.6), 159 (98.6), 134 (67.1) 109 (43.5), 91 (27.0), 55 (32.4).

**Physalin K (8).** Colorless solid (6 mg): [α] $_D$ = -266 (c = 0 .01, MeOH), **UV** (MeOH) λ$_{max}$ nm (log∈): 229 (3.0). **IR**(KBr) ν $_{max}$ cm$^{-1}$: 3407 (OH), 1775 (lactone), 1630 (C = C-C = O), **$^1$H-NMR** (CDCl$_3$, 500 MHz):1.12 (s, H$_3$-28), 1.20 (s, H$_3$-19), 1.90 (s, H$_3$-21), 3.90 (d, $J_{6α, 7β}$ = 3.0, H-6), 4.55 (m, H-22), 3.76 (d, $J_{27,27}$ = 13.5 Hz, H$_2$-27), 4.49 (dd, $J_{22,23α}$ = 13.5 Hz, $J_{22,23β}$ = 5.0 Hz, H-22), 6.65 (dd, $J_{3,2}$ = 6.5 Hz, $J_{3,4}$ = 8.0 Hz, H-3), 7.00 (dd, $J_{4,2}$ = 1.0 Hz, $J_{4,3}$ = 8.5 Hz, H-4), **HRESI-MS** *m/z*: 559.1800 [C$_{28}$H$_{30}$O$_{12}$+H; 559.1810].

*Single-crystal X-ray diffraction data.* Empirical formula = C$_{28}$ H$_{30}$ O$_{12.}$ H$_2$O. CH$_3$OH, Mr = 358.46, Crystal system: Monoclinic, space group: P2$_1$, Unit cell dimensions: **a** = 7.4821 (5)Å, **b** = 11.7099(7)Å, **c** = 15.2299(11)Å, Volume: 1327.52(15)Å$^3$, Z = 2, $ρ_{calc}$ = 1.522 Mg/m3, F(000) 776, Crystal size: 0.270 x 0.130 x 0.010 mm, θ range for data collection: 4.773 to 68.237 deg. A total of 17,151 reflections collected, out of which 4,824 reflections were judged observed (Rint = 0.1337), Final R indices were, R$_1$ = 0.0521for [I>2σ (I)], wR$_2$ = 0.1156, R indices were R$_1$ = 0.0816, wR$_2$ = 0.1300 for all data, largest diff. peak and hole: 0.349and—0.269e. Å$^{-3}$, CCDC No. 2177423.

**Physalin D (9).**   Colorless solid (6 mg): $[\alpha]_D$ = +78.4 (c = 0 .01, MeOH), **UV (MeOH)** $\lambda_{max}$ nm (log∈): 230 (3.2).**IR** (KBr) $\nu_{max}$ cm$^{-1}$: 3377 (OH), 1788 (lactone), 1653 (C = C-C = O),**$^1$H-NMR** (CDCl$_3$, 500 MHz):1.25 (s, H$_3$-28),1.29 (s, H$_3$-28), 1.98 (s, H$_3$-28), 3.76 (m, H-6), 4.53 (m, H-22), 3.74 (m, H$_2$-27), 4.50 (d, $J_{27,25}$ = 5.0 Hz, H$_2$-27), 6.65 (ddd, $J_{3,2}$ = 10.5, Hz, $J_{3,4a}$ = 4.5 Hz, $J_{3,4ß}$ = 2.5, H-3), 5.90 (dd, $J_{2,3}$ = 10.5 Hz, $J_{2,4ß}$ = 2.5 Hz, H-2), **HRE-SI-MS** [M+H]$^+$ *m/z*: 545.2013 [C$_{28}$H$_{32}$O$_{11}$+H; 545.2004].

## Promastigote growth and inhibition assay

*L. major* and *L. tropica* (Clinical isolates) promastigotes were grown at 22 ± 25°C in RPMI-1640 (Sigma) medium containing 10% of heat-inactivated (56°C for 30 min) fetal bovine serum. Cell viability was initially evaluated by wet mount method by observing the live cells under compound microscope. Parasite growth was assessed by counting live / motile promastigotes in a Neubauer chamber. Viable cell count was used for the further experimentations (treatments) [25].

For the estimation of particular concertation at which tested compound caused 50% inhibition (IC$_{50}$) of resistant cell proliferation concerning untreated controls, the MTT assay was employed [26–28]. The promastigotes in their log phase were used in 96-well plates. $1 \times 10^6$ wild-type and resistant promastigotes were dispensed in 96 96-well plates, and incubated with tested compounds at a range concentration of 200–10 μM at 27°C for 72 h. After incubation of 72 h, MTT dye (3-(4,5-dimethylthiazol-2-yl)-2,5-diphenyl-2H-tetrazolium bromide) was added and further incubate it for 3–4 h. Amphotericin B, pantamidine and miltefosine were used as the positive control, untreated promastigotes were used as negative control moreover DMSO control was also added in the study as a stock solution of the compounds that were prepared in DMSO. All the experiments were performed in triplicates. After completion of the experiment the absorbance by Multiskan ascent plate reader at 452 nm and percent inhibition was calculated by the following formula.

Cytotoxic concentration (%) = [100- (Absorbance of test)/ (Absorbance of solvent control) ×100] %.

**MIL-resistant line.**   Promastigote cultures *tL. tropica* (Clinical isolates) were maintained at 26°C in RPMI-1640 medium containing 15% FBS and 1% penicillin–streptomycin mixture.

Generation of miltefosine-unresponsive strain was carried under high MIL pressure by *in vitro* passage with a stepwise increase in the MIL concentration. At each step, parasites were cultured and passaged every 3–4 days at an initial concentration of 5x10$^5$ promastigotes/mL in order to achieve -stable growth as compared to the wildtype isolate. Growth rates were measured for resistant populations and compared with the WT strain. Parasites were counted at an initial concentration of 5x10$^5$ parasites/mL and growth was measured daily using a Neubauer chamber until the population reached the stationary phase. Furthermore, the fluorescence microscopic investigations via DAPI stain were also carried out to supplement the study [29, 30].

**Cytotoxicity against BJ and 3T3 cell lines.**   The cytotoxicity of the test compounds against normal cell lines was evaluated through MTT (3-[4, 5-dimethylthiazole-2-yl]-2, 5-diphenyl-tetrazolium bromide] colorimetric assay in 96-well flat-bottomed microplates [31]. The 3T3 (mouse fibroblast) and BJ (human fibroblast) were cultured Dulbecco's Modified Eagle Medium, supplemented with 5% of fetal bovine serum (FBS), 100 IU/mL of penicillin and 100 μg/mL of streptomycin in 75 cm2 flasks, and then incubated in 5% CO$_2$ incubator at 37°C. The growing cells were counted by using a haemocytometer, and diluted with the particular medium. The 5x10$^4$ cells/mL of the cell culture was prepared, and a sample (100 mL) was introduced into each well. The medium was removed after overnight incubation, and 200 μL of fresh medium was added with different concentrations of compounds (1–30 μM). After 48

hours, 200 μL MTT (0.5 mg/mL) was added, and the incubation was further continued for 4 hrs. After that time, 100 μL of DMSO was added to each well. The absorbance was measured using a micro plate reader (Spectra Max Plus, Molecular Devices, CA, USA) at 570 nm for the extent of MTT reduction to formazan within the cells. The cytotoxicity was recorded as CC50 against cells by using Soft- Max Pro software. Doxorubicin and cycloheximide were used as positive control. The experiments were run in triplicate.

### Docking studies

In order to predict the tentative binding affinities and the possible mode of inhibition *via* these compounds, molecular docking studies against various leishmanial metabolic enzymes that have role in the growth and survival of the parasite were conducted.

**Protein preparation.** For docking studies, the selected protein targets *i. e*., pteridine reductase (PDB:1E92), pyruvate kinase (PDB:3PP7), glyceraldehyde-3-phosphate dehydrogenase (PDB:1I32), phosphoglucose isomerase (PDB:1T10), dihydroortate dehydrogenase (PDB:4EF8), fructose-1,6-bisphosphatase (PDB:5OFU), SAM-dependent methyltransferase (PDB:1XTP), mitochondrial fumarate hydratase (PDB: 6MSO), *N*-myristoyltransferase (PDB: 6QDB), and MAP kinase (PDB: 3UIB), were prepared by *Protein Preparation Wizard* in Maestro Schrödinger 12 [32, 33]. During preparation, the missing hydrogens were added, and partial charges were assigned using OPLS-3e force field. Hydrogens and heavy atoms were optimized by restrained minimization.

**Ligand preparation and database generation.** The 2D structures of compounds were converted to 3D structures, *via* the Ligprep module in Maestro Schrödinger 12 [34]. *Ligprep* was used to correct the protonation, and ionization states of the compounds, and assigned proper bond orders. Afterwards, the tautomeric and ionization states were created for each ligand.

**Receptor grid generation and molecular docking.** The grid box was defined by selecting the co-crystallized molecules in the binding site of abovementioned selected protein targets to keep the center of each docked ligand with the same dimensions of binding box. Rigid receptor docking protocol was run in standard precision (SP) mode of Glide based on OPLS-3e force field [35–37]. During the process of docking, the protein was fixed, while ligands were flexible.

The molecular mechanics-generalized Born surface area (MM-GBSA) method in Prime was used for rescoring the docked pose of ligand in the binding site of the selected target protein [38]. These poses were taken as inputs for the energy minimization of the protein-ligand complexes ($E_{complex}$), free protein ($E_{protein}$), and free ligands ($E_{ligand}$). The binding free energy $\Delta G_{bind}$ was determined according to the following equation:

$$\Delta Gbind = Ecomplex\ (minimized) - Eligand\ (minimized) - Ereceptor\ (minimized)$$

**Statistical analysis.** Three replicates were used in each experiment, unless otherwise stated. All results were presented as means standard deviations. A one-way ANOVA was used to analyze statistically differences at a *P*-value 0.05 (95% confidence interval) in conjugation with the Dunnett test using Graph pad prism version 9.4.1 (California. San Diego).

## Results

### Isolation and structural characterization

The whole plant of *P. minima* Linn. yielded nine compounds **1–9**, including a new aminophysalin (**1**). These compounds were characterized by spectroscopic techniques (Fig 1).

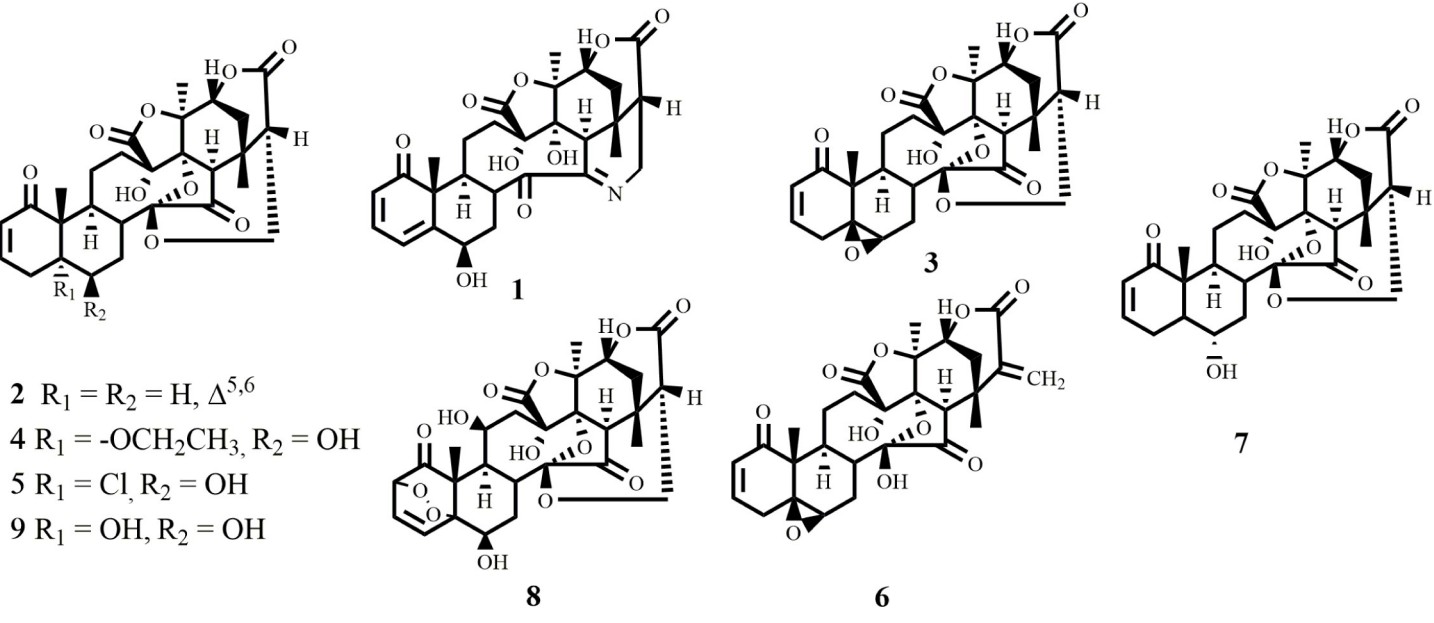

**Fig 1. The structures of compounds 1–9 isolated from *Physalis minima*.**

Compound **1** was obtained as a colorless crystalline solid. The ESI-MS of 1 exhibited the $[M+H]^+$ at *m/z*: 526.2059 [$C_{28}H_{32}NO_9$+H; 526.3072]. The IR spectrum showed a strong absorption at 3463 cm$^{-1}$, indicating the presence of OH groups. The absorption band at 1732 cm$^{-1}$ is a characteristic absorption of physalin, which appeared due to a five-membered lactone ring. Furthermore, the IR absorption at 1656 cm$^{-1}$ was assigned to an α,β-unsaturated ketone [39–41]. The $^1$H-NMR spectrum (Table 1) of compound **1** showed three characteristic methyl singlets, resonated at δ 1.54, 1.85, and 1.04 and assigned to H-19, H-21, and H-28 methyl respectively. The appearance of three mutually coupled olefin carbon signals appeared at δ 6.00 (d, $J_{2,3}$ = 10.0 Hz), 6.98 (dd, $J_{3,4}$ = 9.5 Hz, $J_{3,4}$ = 6.0 Hz) and 6.12 (dd, $J_{4,3}$ = 5.5 Hz) indicated a 2-ene-1-one system in ring A. The presence of nitrogen-containing bridge between C-15, and C-27 (*i.e* entirely different from other physalin) was inferred from the appearance of a pair of characteristic signals δ 4.17 (dd, $J_{27,25}$ = 20.5 Hz, $J_{27,27}$ = 10.0 Hz) and δ 4.46 ($J_{27,27}$ = 21) for H$_2$-27 [42]. An overlapped signal also appeared at δ 4.49 indicating the hydroxylation at C-6 carbon [43]. $^{13}$C-NMR spectra (Table 1) showed 28 carbon signals, assigned to eleven quaternary, nine methine, five methylene, and three methyl carbons. There were four carbon signals with chemical shift above δ 170, and assigned to two ketone carbonyls [$δ_c$ 210.7 (C-1), $δ_c$ 208.8 (C-14)] and two ester carbonyls [$δ_c$ 171.3 (C-18), $δ_c$ 173.0 (C-26)]. The presence of an additional signal at δ 164.9 (C-15), and downfield shift of C-27 at δ 47.7 suggesting the nitrogen-containing bridge between C-15 and C-27. The presence of carbonyl carbon at C-14 was supported by key HMBC correlations of δ 3.13 (H-8) and 2.52 (H-16) with $δ_c$ 208.8 (C-14) and 164.9 (C-15), respectively (Fig 2). The HMBC correlation of H$_2$-27 (δ 4.46, 4.17) with 164.9 (C-15) supported the nitrogen-containing bridge. COSY correlation between H-25 and H-27 was also supporting the proposed structure. The OH at C-6 was deduced based on HMBC correlations of H-4 (δ 6.12) with C-6 (δ 70.4) (Fig 2).

The structure of compound **1** was unambiguously deduced based on single-crystal X-ray diffraction analysis (Figs 4 and 5). The ORTEP diagram depicted that the molecule consists of seven fused rings, *i.e.* rings A (C1-C5/C10), B (C5-C10), C(C8/C9/C11-17), D (C13/C17/C18/C20/O9/O10), E (C16/C17/C20/C22-C24), F (C22-C26/O-9), and G (C-15/C-16/C-24/

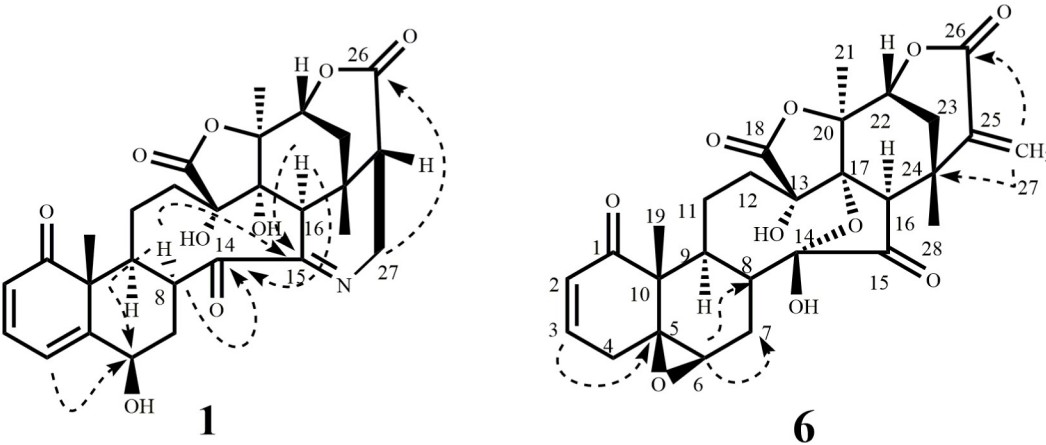

**Fig 2. Key COSY (──) and HMBC (---→) correlations in compounds 1 and 6.**

C27-N1). Rings A and B transfused with each other through C5/C10 linkage and exists in half chair and chair conformations, respectively. Ring C (C8/C9/C11-17) appeared in twisted-boat-chair conformation with the cleavage of ether linkage between rings D and E as appeared in other reported physalins. Tetrahydropyridine moiety *i.e.* ring G (C15/C16/C24/C27N1) is completely different from reported physalin [39, 40]. The result showed that C6-OH and 28-CH₃ were on the same side and assigned to be β-oriented. Whereas C13-OH, C17-OH and C21-CH₃ were deduced to be α-oriented (Fig 3). In the crystal lattice molecules found to be linked to form a three-dimensional network (Fig 4). Previously reported, aminophysalin A [42] was the first physalin having an unusual structural feature in with a nitrogen atom. Based on spectroscopic compound **1** was s found to be the second member of aminophysalin series.

Compounds **2–5** were identified as previously reported physalin B (**2**) [44, 45], 5β, 6β-epoxyphysalin B (**3**) [45], 5α-ethoxy-6ß-hydroxy-5,6-dihydrophysalin B (**4**) [42, 46], and physalin H (**5**) [47] by comparison of the reported spectroscopic data. These compounds had first reported from *P. angulata*, *P. alkekengi*, and *P. minima*, respectively.

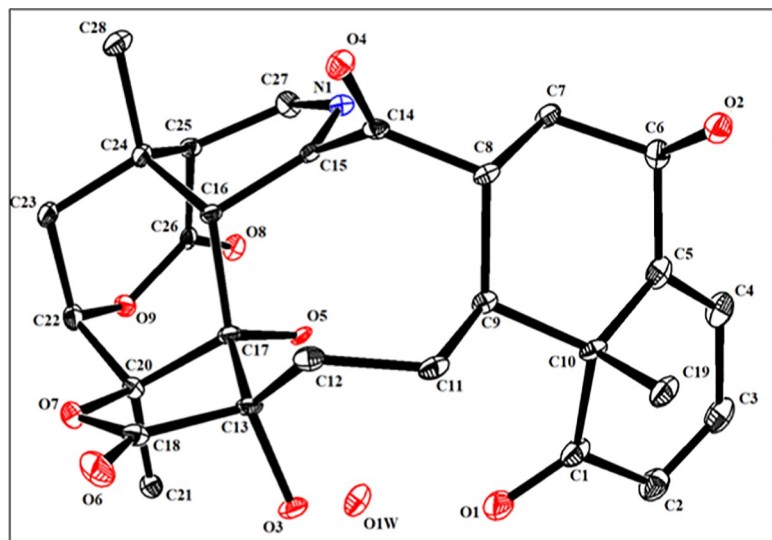

**Fig 3. ORTEP view of compound 1 at 50% probability level.**

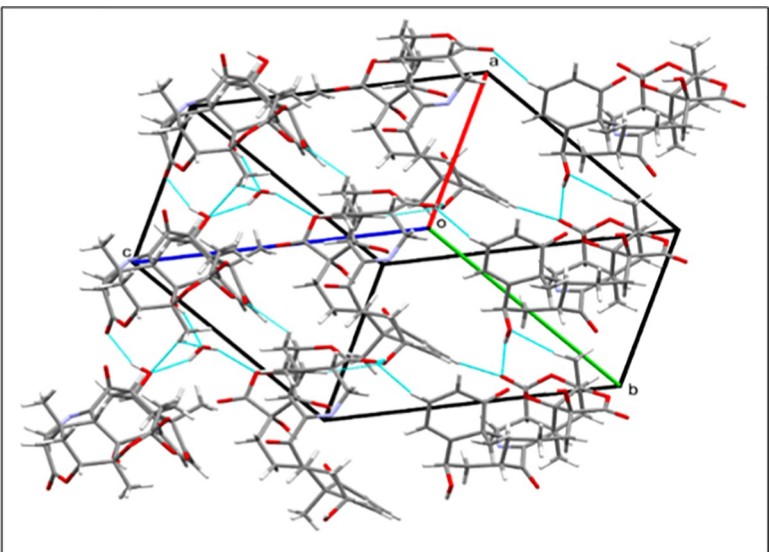

**Fig 4. Crystal packing diagram of compound 1.**

Compound **6** was isolated here first time from *Physalis minima*. Its structure was previously reported only on a tentative assignment based on the fragmentation pattern studied through UHPLC-QTOF-MS/MS analysis *Physalis alkekengi L. var. franchetii* (Mast.) [41]. Therefore, this is the first detailed report of the isolation and validated structure characterization of 5ß,6ß-epoxyphysalin C (**6**). Compound **6** was obtained as a white solid. The ESI-MS [M +H]$^+$m/z: 527.1907 [$C_{28}H_{32}NO_9$+H; 527.1912]. The IR spectrum showed strong absorption at 3427, 1782, and 1656 indicating the presence of OH groups, five-membered lactone ring and α,β-unsaturated ketone, respectively. The $^1$H-NMR spectrum (Table 1) showed three characteristic methyl singlets at δ1.21, 1.92, and 1.58, and assigned to $H_3$-19, $H_3$-21, and $H_3$-28 methyl respectively. Two mutually coupled olefin proton signals, appeared at δ 5.99 (dd, $J_{2,4}$ = 10.0 Hz, $J_{2,3}$ = 2.5 Hz) and δ 6.83 (ddd, $J_{2,3}$ = 3.0 Hz, $J_{3,4}$ = 6.0 Hz, $J_{3,4}$ = 10.0 Hz) indicated a 2-ene-1-one system in ring A. A one-proton downfield broad singlet at δ 3.24 (d, *J* = 3.0 Hz) was assigned to the C-6 methine proton, geminal to oxy group. The assigned C-6 proton at δ 3.24 was further supported by HMBC correlations with C-8 (δ 39.1) and C-7 (δ 26.4) (Fig 2).

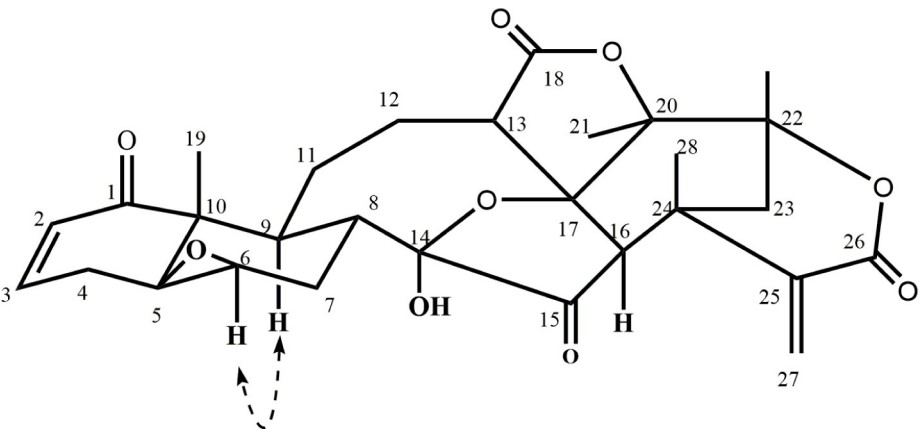

**Fig 5. Key NOESY correlations in compounds 6.**

HMBC correlation of H-3 (δ 6.83) with C-5 (δ 61.8) supported the presence of an epoxide between C-5 and C-6. The low *J* value (*J* = 3.0 Hz) and comparison of the chemical shift with a known physalin (5β,6β-epoxyphysalin B) suggested α-orientation of H-6 [32]. Two characteristic singlets at δ 6.64 (s) and 5.72 (s) indicating the absence of oxygen containing bridge between C-14 and C-27. An olefin at C-27 was inferred by HMBC correlations of $H_2$-27 (δ 6.64, δ 5.72) with C-26 (δ 163.7) and C-24 (δ 36.6). The $^{13}$C-NMR data (Table 1) showed 28 carbon signals assigned to twelve quaternary, seven methine, six methylene, and three methyl carbons. The signal at δ 64.4 was assigned to C-5, and indicated the epoxidation at C-6. Two additional quaternary signals at δ 102.1 and 139.3 were assigned to C-14 and C-24, respectively. NOESY correlation between the H-9 (δ 2.51) and H-6 (δ 3.24) indicated a β epoxide (Fig 5). Compound **6** was identified as 5β, 6β-epoxyphysalin C based on spectroscopic analysis.

Compounds **7** and **9** were identified as known physalins G (**7**) and physalin D (**9**) through comparison of their reported spectroscopic data [43, 48].

The first report for the isolation of physalin K (**8**) was from *Physalis minima* in 1980 [43]. However the structure was extensively revised later based on spectroscopic techniques in 1995 [49]. We have studied the structural parameters of physalin K (**8**) based on single-crystal X-ray diffraction data and found it in good agreement with the revised structure.

Like other physalin, the molecular skeleton was found to have a rigid framework, consisting of eight condensed rings. Two six-membered rings *i.e.* rings A (C1/C2/C5/C10/O2/O3) and B (C2-C5/O2/O3) linked together through epi-dioxy linkage at C2 and C5 adopts screw boat conformation. Ring C (C5-C10) is transfused with ring A through C5/C10 bonds, and exists in chair conformation. The eight-membered ring D (O7/C8/C9/C11-C13/C17/C14) and adopts a boat-chair conformation, connected with two *spiro*-fused rings E (C13/C17/C18/C20/O9/ O10) and F (C14-C17/O7) through O7/C13/C14/C17- /O7, while dimethyl substituted six-membered ring G (C-16/C-17/C-22—C-24) is connected with two lactone ring moieties of H (C14-C16/C24-C27/O6) and I (C22-C26/O11/O12) along C16/C23-C25 linkage (Figs 6 and 7).

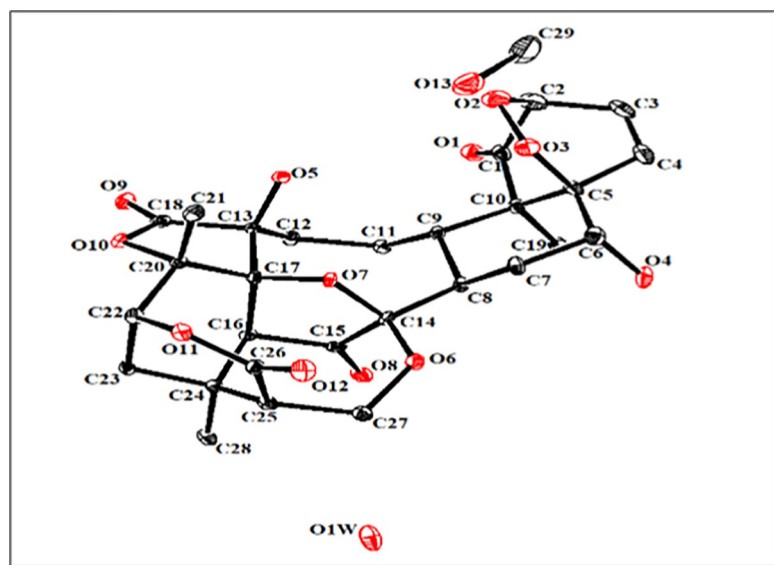

**Fig 6. *ORTEP* view of compound 8 at 50% probability level.**

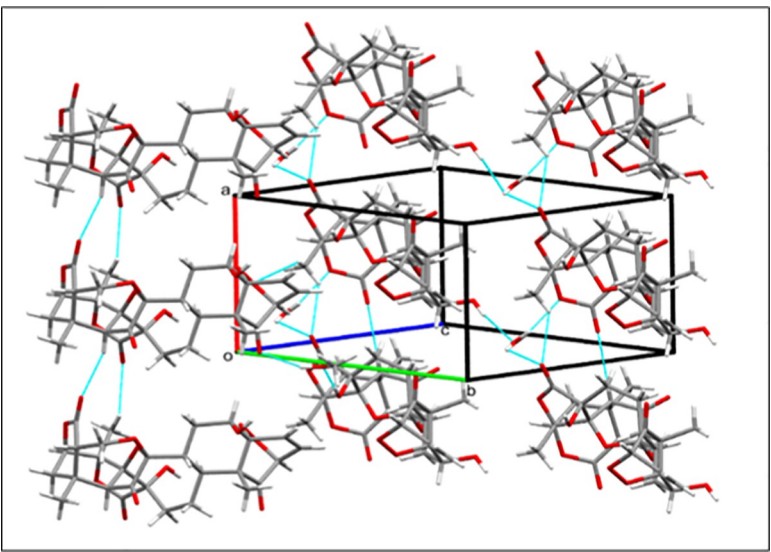

**Fig 7. Crystal packing diagram of compound 8.**

## Anti-leishmanial activity

In order to evaluate the anti-leishmania activity, *Physalis minima* extracts (Ethanol/water, Pet ether, Dichloromethane, and Ethyl acetate), active fractions (A-D), and pure physalins **1–9** were incubated in various concentrations with axenic cultures of *L. tropica* and *L. major* in comparison to three commercially available drugs, amphotericin B, pentamidine, and miltefosine (Tables 2 and 3). These drugs were used as a positive control to broaden the range due to different mode of actions. The $IC_{50}$ values (Table 3) indicated that chloro substituted physalin H (**5**) was the most active member ($IC_{50}$ = 9.59 ± 0.27 μM) (*P-value* = 0.0009 for pantamidine while for miltefosine and amphotericin B the *P-value* was = < 0.0001), followed by physalins B (**2**, $IC_{50}$ = 13.33 ± 0.098 μM), physalins G (**7**, $IC_{50}$ = 19.49 ± 0.02 μM), and 5ß,6ß-epoxyphysalin C (6, $IC_{50}$ = 23.76 ± 1.10 μM) (*P-value* = < 0.0001 for all three tested standards)against *L. tropica* and showed a more potent activity than the standards pentamidine ($IC_{50}$ = 27.20 ± 0.01 μM) and miltefosine ($IC_{50}$ = 42.75 ± 1.03 μM). The anti-leishmanial activity of

**Table 2. Anti-leishmanial activity of extracts and fractions against *L. major* and *L. tropica*.**

| | | *L. major* | | | | *L. tropica* | | | |
|---|---|---|---|---|---|---|---|---|---|
| | | IC$_{50}$ (μg/ml ± SD) | p-value Amphotericin | p-value Pentamidine | p-value Miltefosine | IC$_{50}$ (μg/ml ± SD) | p-value Amphotericin | p-value Pentamidine | p-value Miltefosine |
| Extracts | Ethanol/water | >100 | - | | | >100 | - | | |
| | Pet ether | >100 | - | | | >100 | - | | |
| | Dichloromethane | 19.01±0.7 | <0.0001 | <0.0001 | <0.0001 | 19.53±1.3 | <0.0001 | <0.0001 | 0.0021 |
| | Ethyl acetate | >100 | - | | | >100 | - | | |
| Fractions | A | 6.67±0.1 | <0.0001 | <0.0001 | <0.0001 | 10.16±2.0 | <0.0001 | 0.0568 | <0.0001 |
| | B | 9.95±1.9 | <0.0001 | 0.8898 | 0.5084 | 6.41±0.074 | <0.0001 | 0.0001 | <0.0001 |
| | C | 6.30±0.018 | <0.0001 | <0.0001 | <0.0001 | 6.34±0.045 | <0.0001 | <0.0001 | <0.0001 |
| | D | 6.37±0.015 | <0.0001 | <0.0001 | <0.0001 | 6.37±0.015 | <0.0001 | 0.0001 | <0.0001 |
| Standard drugs | Amphotericin | 3.14 ± 0.04 | | | | 3.16 ± 0.04 | | | |
| | Pentamidine | 9.25 ± 0.005 | | | | 9.25 ± 0.005 | | | |
| | Miltefosine | 10.4 ± 0.42 | | | | 17.4 ± 0.42 | | | |

**Table 3. Anti-leishmanial activity of compounds 1–9 against *L. major* and *L. tropica*.**

| Compounds | *L. major* | | | | *L. tropica* | | | |
|---|---|---|---|---|---|---|---|---|
| | IC$_{50}$ (µM± SD) | p-value Amphotericin | p-value Pentamidine | p-value Miltefosine | IC$_{50}$ (µM± SD) | p-value Amphotericin | p-value Pentamidine | p-value Miltefosine |
| 1 | Inactive | - | - | - | Inactive | - | - | - |
| 2 | 3.04 ± 1.12 | 0.9001 | <0.0001 | <0.0001 | 13.33 ± 0.098 | <0.0001 | <0.0001 | <0.0001 |
| 3 | 3.76 ± 0.85 | 0.9997 | <0.0001 | <0.0001 | 18.53 ± 0.28 | <0.0001 | <0.0001 | <0.0001 |
| 4 | Inactive | 0.9999 | <0.0001 | <0.0001 | Inactive | <0.0001 | <0.0001 | <0.0001 |
| 5 | 3.34 ± 0.64 | <0.0001 | <0.0001 | <0.0001 | 9.59 ± 0.27 | <0.0001 | 0.0009 | <0.0001 |
| 6 | 84.79 ± 1.56 | <0.0001 | <0.0001 | <0.0001 | 23.76 ± 1.10 | <0.0001 | <0.0001 | <0.0001 |
| 7 | 52.28 ± 1.18 | 0.9001 | <0.0001 | <0.0001 | 19.49 ± 0.02 | <0.0001 | <0.0001 | <0.0001 |
| 8 | inactive | - | - | - | inactive | - | - | - |
| 9 | inactive | - | - | - | 55.24± 0.01 | <0.0001 | <0.0001 | <0.0001 |
| Amphotericin B | 3.39 ± 0.043 | | | | 3.42 ± 0.04 | | | |
| Pentamidine | 27.20 ± 0.015 | | | | 27.20 ± 0.01 | | | |
| Miltefosine | 25.55 ± 1.03 | | | | 42.75 ± ± 1.03 | | | |

physalins B (**2**), 5ß,6ß-epoxyphysalin B (**3**) and physalin H (**5**) was found in agreement with previous data reported by us [23] against *L. major*, in comparison to amphotericin B. In the current study, the comparison of the anti-leishmanial activity of physalins B (**2**), 5ß,6ß-epoxy-physalin B (**3**) and physalin H (**5**) against *L. major* include comparison with pentamidine and miltefosine. The results indicate that growth inhibition potential of physalins B (**2**, IC$_{50}$ = 3.04 ± 1.12 µM) (*P-value* = 0.9001 for amphotericin B while for miltefosine and pantamidine the *P-value* was = < 0.0001), 5ß,6ß-epoxyphysalin B (**3**, IC$_{50}$ = 3.76 ± 0.85 µM) (*P-value* = 0.9997 for amphotericin B while for miltefosine and pantamidine the *P-values* was = < 0.0001) and physalin H (**5**, IC$_{50}$ = 3.34 ± 0.64 µM) (*P-value* for amphotericin B, miltefosine and panta-midine the *P-value* was = < 0.0001). These were several-fold more potent than the standard drugs pentamidine (IC$_{50}$ = 27.20 ± 0.015 µM) and miltefosine (IC$_{50}$ = 25.55 ± 1.03 µM). Com-pounds **1**, **4,** and **8** appeared to be inactive against both *L. major* and *L. tropica* strains (Table 3). Compounds **3**, **5**, and **6** showed low selectivity index (SI) values of 0.19, 0.119, and 0.007, respectively against *L. major* tested on a 3T3 cell line. However, compound **2** showed SI value of 3.28. All these compounds showed <1 SI index against *L. tropica* cultured on 3T3 cells. Moreover, on the human fibroblast cell line (BJ), compounds **2**, **3**, **5**, and **6** showed ≥1 SI values of 3.5, 7.34, 3.11, and 1.19 against *L. major*, respectively. While compounds **3**, **5**, and **6** also showed SI values of 1.48, 1.08, and 4.20 against *L. tropica*. The SI value higher than 1 indi-cated that these compounds are theoretically more effective and safe drugs. However, com-pound 2 demonstrated a SI value of 0.8 on *L. tropica*. (Tables 4–7).

## Anti-leishmanial activity against MIL-resistant *L. tropica*

The safe, non-toxic and effective drug, miltefosine was selected after evaluating its cytotoxicity on BJ cell line. The selection of miltefosine depends upon its potential for the treatment of Leishmaniasis along with the rapid acquisition of resistance. Several studies have reported mil-tefosine for the treatment of *L. tropia*. It also works well during the study presented against locally isolated strains of *L. tropica*. Furthermore, compounds **2–9** were also evaluated for their potential against the Miltefosine-unresponsive strain (MIL resistant) *L. tropica*. Compounds **4** (IC$_{50}$ = 74.65 ± 0.81 µM) and **7** (IC$_{50}$ = 39.44 ± 0.65 µM) (*P-value* was = < 0.0001) were found to be more potent anti-leishmanial agents against the MIL resistant promastigotes than the

**Table 4. Selectivity index of pure compounds against BJ fibroblast cell line.**

| | Samples | BJ cell line | *L. major* | *L. tropica* | *L. major* | *L. tropica* |
|---|---|---|---|---|---|---|
| | | CC50 (µM± SD) | IC$_{50}$ (µM± SD) | IC$_{50}$ (µM± SD) | SI (CC$_{50}$/ IC$_{50}$) | SI (CC$_{50}$/ IC$_{50}$) |
| **Extracts** | Ethanol/water | >100 | >100 | >100 | - | - |
| | Pet ether | >100 | >100 | >100 | - | - |
| | Dichloromethane | 26.5±1.1 | 19.01±0.7 | 19.53±1.3 | 1.39 | 1.35 |
| | Ethyl acetate | >100 | >100 | >100 | - | - |
| **Fractions** | Fraction-A | 10.4 ± 1.0 | 6.67±0.1 | 10.16±2.0 | 1.49 | 1.02 |
| | Fraction-B | >100 | 9.95±1.9 | 6.41±0.074 | - | - |
| | Fraction-C | >100 | 6.30±0.018 | 6.34±0.045 | - | - |
| | Fraction-D | >100 | 6.37±0.015 | 6.37±0.015 | - | - |

**Table 5. Selectivity index of crude fractions against BJ fibroblast cell line.**

| Samples | BJ cell line | *L. major* | *L. tropica* | *L. major* | *L. tropica* |
|---|---|---|---|---|---|
| | CC50 (µM± SD) | IC$_{50}$ (µM± SD) | IC$_{50}$ (µM± SD) | SI (CC$_{50}$/ IC$_{50}$) | SI (CC$_{50}$/ IC$_{50}$) |
| **1** | Non cytotoxic | Inactive | Inactive | - | |
| **2** | 10.8 ± 1.5 | 3.04 ± 1.12 | 13.33 ± 0.098 | 3.5 | 0.8 |
| **3** | 27.6 | 3.76 ± 0.85 | 18.53 ± 0.28 | 7.34 | 1.48 |
| **4** | 14.2 ± 0.8 | Inactive | Inactive | - | - |
| **5** | 10.4 ± 1.0 | 3.34 ± 0.64 | 9.59 ± 0.27 | 3.11 | 1.08 |
| **6** | >100 | 84.79 ± 1.56 | 23.76 ± 1.10 | 1.19 | 4.20 |
| **7** | >100 | 52.28 ± 1.18 | 19.49 ± 0.02 | - | - |
| **8** | >100 | Inactive | Inactive | - | - |
| **9** | >100 | Inactive | 55.24± 0.01 | - | - |
| **Amphotericin** | 16.059 | 3.39 ± 0.043 | 3.42 ± 0.04 | 4.37 | 4.69 |

**Table 6. Selectivity index of pure compounds against 3T3 mouse fibroblast cell line.**

| Samples | 3T3 Cell line | *L. major* | *L. tropica* | *L. major* | *L. tropica* |
|---|---|---|---|---|---|
| | CC50 (µM± SD) | IC$_{50}$ (µM± SD) | IC$_{50}$ (µM± SD) | SI (CC$_{50}$/ IC$_{50}$) | SI (CC$_{50}$/ IC$_{50}$) |
| **1** | - | Inactive | Inactive | - | - |
| **2** | 10 ± 0.4 | 3.04 ± 1.12 | 13.33 ± 0.098 | 3.28 | 0.007 |
| **3** | 0.72 ± 0.5 | 3.76 ± 0.85 | 18.53 ± 0.28 | 0.19 | 0.038 |
| **4** | 19.6± 0.8 | Inactive | Inactive | - | - |
| **5** | 0.4 ± 0.07 | 3.34 ± 0.64 | 9.59 ± 0.27 | 0.119 | 0.041 |
| **6** | 0.6 ± 0.04 | 84.79 ± 1.56 | 23.76 ± 1.10 | 0.007 | 0.025 |
| **7** | Non cytotoxic | 52.28 ± 1.18 | 19.49 ± 0.02 | - | - |
| **8** | Non cytotoxic | Inactive | Inactive | - | - |
| **9** | Non cytotoxic | Inactive | 55.24± 0.01 | - | - |

standard Miltefosine (IC$_{50}$ = 169.55 ± 0.78 µM). It is interesting to be noted that compound **4** found to be inactive against clinical isolates of *L. major* and *L. tropica* (Table 8). Any possible alteration in nucleus and cytoplasm caused by the acquisition of MIL resistance was evaluated by DAPI staining. No morphological changes or any other distortion in growth of resistance lines were observed after microscopic observations and supported the propagation and survival of *Leishmania* parasite in higher concentration of drugs. Collectively, the microscopic

**Table 7. Selectivity index of pure compounds against 3T3 mouse fibroblast cell line.**

| | Samples | 3T3 cell line | *L. major* | *L. tropica* | *L. major* | *L. tropica* |
|---|---|---|---|---|---|---|
| | | CC50 (μM± SD) | IC$_{50}$ (μM± SD) | IC$_{50}$ (μM± SD) | SI (CC$_{50}$/ IC$_{50}$) | SI (CC$_{50}$/ IC$_{50}$) |
| **Extracts** | Ethanol/water | >100 | >100 | >100 | - | - |
| | Pet ether | >100 | >100 | >100 | - | - |
| | Dichloromethane | 4.5 ± 0.5 | 19.01±0.7 | 19.53±1.3 | 0.23 | 0.23 |
| | Ethyl acetate | >100 | >100 | >100 | - | - |
| **Fractions** | Fraction-A | 12.2 ± 1.5 | 6.67±0.1 | 10.16±2.0 | 7.30 | 1.20 |
| | Fraction-B | 4.8 ± 1.6 | 9.95±1.9 | 6.41±0.074 | 0.48 | 0.74 |
| | Fraction-C | 2.7 ± 0.3 | 6.30±0.018 | 6.34±0.045 | 0.42 | 0.42 |
| | Fraction-D | 3.3 ± 0.3 | 6.37±0.015 | 6.37±0.015 | 0.51 | 0.51 |

**Table 8. Anti-leishmanial activity of compounds 2–9 against the MIL-resistant *L. tropica*.**

| | | MIL Resistant |
|---|---|---|
| | | *L. tropica* |
| | Compounds | IC$_{50}$ (μM± SD) |
| **Pure Compounds** | 2 | inactive |
| | 3 | inactive |
| | 4 | 74.65 ± 0.81 |
| | 5 | inactive |
| | 6 | inactive |
| | 7 | 39.44 ± 0.65 |
| | 8 | inactive |
| | 9 | inactive |
| **Standard drug** | Miltefosine | 169.55 ± 0.78 |

observations based on DAPI staining strongly supported that the viability and dynamics of resistant line were similar to that of wild type.

## Cytotoxicity of physalin against BJ (Human fibroblast) cell lines

Cytotoxicity *Physalis minima* extracts (Ethanol/water, Pet ether, Dichloromethane, and Ethyl acetate), active fractions (A-D), and pure physalins **2–9** were evaluated against normal fibroblast (3T3) and BJ (human fibroblast) cells lines, and all tested compounds were found to be non-cytotoxic in nature (Tables 4 and 5).

## Docking studies

Compounds with significant *in vitro* anti-leishmanial activities proceeded for docking studies against various therapeutically important enzymes involved in different metabolic pathways For this purpose, various enzymes related to major metabolic pathways including glycolysis (pyruvate kinase, glyceraldehyde-3-phosphate dehydrogenase, phosphoglucose isomerase), folate pathway (pteridine reductase), gluconeogenesis (fructose-1,6-bisphosphatase cytosolic and mitochondrial fumarate hydratase), pyrimidine biosynthesis (dihydroortate dehydrogenase), lipid metabolism (*N*-myrositoyltransferase), posttranslational modification (SAM-dependent methyltransferase), and cell signalling (glycogen synthase kinase, MAP kinase) were selected. Among listed targeted enzymes, compounds **2**, **3**, **5**, **6**, **7**, and **9** showed the best binding interaction with phosphoglucose isomerase and *N*-myrositoyltransferase. The docking

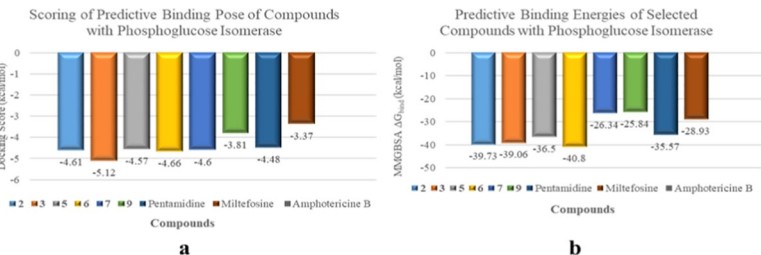

**Fig 8. Docking scores of selected compounds (2, 3, 5, 6, 7, and 9) against leishmanial phosphoglucose isomerase (carbohydrate metabolism).**

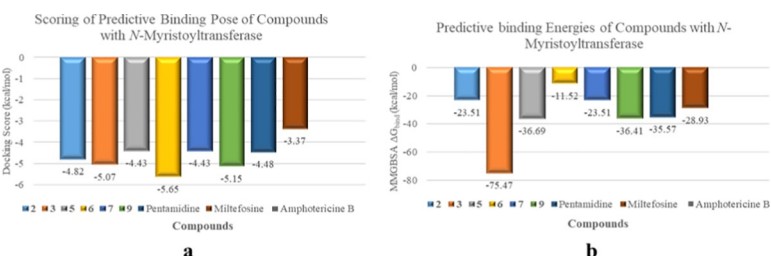

**Fig 9. Docking scores of selected compounds 2, 3, 5, 6, 7, and 9 against leishmanial *N*-myristoyltransferase (lipid metabolism).**

scores were in the range of -4.3 to -5.7 Kcal/mol (Fig 8), while the binding energies were in the range of -11 to -75 Kcal/mol (Fig 9).

## Discussion

In last few decades, extensive research on medicinal plants has contributed to increased of natural products against health disorders globally. In continuation of our work to explore the biological properties of medicinal plants, our group has previously reported the *in vitro* antileishmanial effect of physalins B (**2**), 5ß,6ß-epoxyphysalin B (**3**) and physalin H (**5**), and other isolated physalins from -*Physalis minima* against promastigotes of *L. major* [21, 22], in comparison to amphotericin B. Amphotericin B is the drug of choice for the treatment of various parasitic and fungal infections, including leishmaniasis. However, the clinical use of amphotericin B is limited because of systematic toxicities, and lack of accessibility and affordability [50] The recommended least toxic, non-conventional amphotericin B formulations, such as Fungizone®, AmBisome®, Abelcet®, and Amphocil® are expensive, and, therefore, out of the reach of the poor patients [51, 52]. Unfortunately, the first line pentavalent antimonials are also not available in many leishmanial endemic countries. Therefore, alternatives such as pentamidine, and most recently miltefosine are the recommended US-FDA approved drugs, prescribed by dermatologists for CL. One of the recommended treatment regimens for leishmaniasis includes the combination of amphotericin B and miltefosine. A study conducted in Ethiopia, one of the CL which is caused by *L. aethiopica*, was found to be more severe and difficult to treat as compared to CL caused which is caused by other species. As far as treatment options are concerned, Miltefosine is the only oral anti-leishmanial drug, with a favourable side-effect profile compared to routinely available sodium stibogluconate (SSG) [53, 54]. Therefore, in the current study of the inhibition abilities of isolated physalins towards promastigotes of *L. major* and *L. tropica* were studied *in vitro*, and compared with amphotericin B, pentamidine, and miltefosine. The physalins B (**2**), 5ß,6ß-epoxyphysalin B (**3**), and chloro

substituted Physalin H (**5**) were previously reported by us as -promising candidates with significant abilities to reduce the growth of promastigotes of *L. major*, in comparison to amphoterin B [21]. The published results were further validated in the present study and we further conclude that in addition to amphotericin B, these tested compounds also have several-fold more inhibition potential against promastigotes of *L. major* than pentamidine (IC$_{50}$ = 27.20 ± 0.015 μM), and miltefosine (IC$_{50}$ = 25.55 ± 1.03 μM). In the current study, the test compound also showed potently anti-leishmanial abilities against *L. tropica* promatigotes *in vitro*, and appeared to be several fold more active than tested standards pentamidine (IC$_{50}$ = 27.20 ± 0.01 μM), and miltefosine (IC$_{50}$ = 42.75 ± 1.03 μM). The activity of some selected compounds was also evaluated against a lab generated MIL-resistant line. Compound **2** showed SI value of >1 while compounds **3**, **5**, and **6** showed <1 SI against *L. major* on 3T3 cell line. Compounds **3**, **5**, and **6** showed >1 SI values against both *L. major* and *L. tropica* cultured on BJ cell line. However, compound **2** demonstrated >1 SI value only against *L. tropica*.

The alkylphosphocholine drug, miltefosine possesses its potential against several parasitic species and different cancer cells, along with activity against various pathogenic fungi and bacteria [55]. Knowledge of experimental MIL resistance in *Leishmania* is limited to defects in drug internalization (defective inward translocation of MIL) and increased drug efflux [56]. A major potential drawback in the use of miltefosine for the treatment of leishmaniasis could be the relatively rapid generation of drug resistance *in vitro* [57]. Miltefosine is considered a breakthrough in the treatment, making it feasible to eliminate a regional disease, unfortunately the acquisition of resistance is of major concern [58]. Mechanisms that are responsible for the resistance acquisition in leishmaina parasite against miltefosine includes reduction in drug uptake, increased efflux and alteration in permeability of the plasma membrane [59–63]. Due to this type of data and research highlights there is a necessity to find out the alternative therapeutic options for leishmaniasis. During the study presented the resistant strain was generated and libraries of compounds were evaluated against the developed line. *L. tropica* MIL-unresponsive / resistant parasites were generated by using the step-wise selection of the drug-miltefosine up to the concentration of 196 μM. No marked difference in growth pattern were also analyzed between WT and MIL-resistant strain. Compounds **4** and **7** were observed as significantly potent as compared to the miltefosine, revealing the potential of these natural compounds against drug–unresponsive strains. However, these results of the current study demonstrated the promising anti-leishmanial potential of physalins against *L. major* and *L. tropica* promastigotes that need to be further validated by *in vivo* studies as an anti-leishamanial agent for oral use.

Targeting the metabolic and biochemical pathways of *Leishmania* is one of the most appropriate therapeutic strategy. Following this rationale, compounds **2**, **3**, **5–7**, and **9** with significant *in-vitro* anti-leishmanial activity were validated through docking studies against various metabolically important enzymes, in order to predict their mechanism of action. Compounds **2**, **3**, **5–7**, and **9** showed better binding affinities against phosphoglucose isomerase and myrositoyltransferase with docking scores in the range of -2 to -7 Kcal, while the binding energies were in the range of -11 to -75 Kcal/mol. The results were then compared with the docking scores and binding energies of anti-leishmanial clinical drugs including amphotericin B, pentamidine, and miltefosin, and found to be comparable with them. The results further indicated that the compounds might affect the glycolytic and lipid metabolic pathways by targeting phospoglucose isomerase and *N*-myristoyltransferase, respectively. Phosphoglucose isomerase (PGI) is considered a promising target for the development of anti-parasitic drugs, as it acts on two essential metabolic pathways, glycolysis and gluconeogenesis. It is an aldose-ketose isomerase that catalyzes the reversible interconversion of G6P into fructose-6-phosphate (F6P) [64].

## Conclusion

Leishmaniasis, a neglected tropical disease is a rapidly growing infection in more than 98 countries of the world. The presented study concluded with the identification of natural steroidal lactones i.e. physalins as potential candidates to be explored further against CL. The complex structures of all natural compounds **1–9** were elucidated *via* combined use of MS, IR, and NMR spectroscopic techniques. The assigned structures of compounds **1**, and **8** were further supported by single-crystal X-ray diffraction studies. Anti-leishmanial activities of isolated physlains were evaluated against *L. major* and *L. tropica* . The literature survey revealed that all of the physalins possess—anti-leishmanial activity against *L. tropica* (clinical isolate of Pakistan), reported for the first time. The promising results of compounds 4 and 7 against the miltefosine-unresponsive *L. tropica* strain (MIL resistant) concluded the anti-leishmanial potential of tested compounds against resistant strain. The compounds were also able to interact with therapeutically relevant enzymes of leishmania including phosphoglucose isomerase, and *N*-myrositoyltransferase.

## Supporting information

**S1 Fig. EI-MS of compound 1.**
(PDF)

**S2 Fig. HREI-MS of Compound 1.**
(PDF)

**S3 Fig. 1H-NMR spectrum of compound 1.**
(PDF)

**S4 Fig. 13C-BB spectrum of compound 1.**
(PDF)

**S5 Fig. 13C-DEPT 135 spectrum of compound 1.**
(PDF)

**S6 Fig. 13C-DEPT 90 spectrum of compound 1.**
(PDF)

**S7 Fig. COSY spectrum of compound 1.**
(PDF)

**S8 Fig. NOESY spectrum of compound 1.**
(PDF)

**S9 Fig. HMBC spectrum of compound 1.**
(PDF)

**S10 Fig. HSQC spectrum of compound 1.**
(PDF)

**S11 Fig. EI-MS of compound 6.**
(PDF)

**S12 Fig. HREI-MS of compound 6.**
(PDF)

**S13 Fig. 1H-NMR spectrum of compound 6.**
(PDF)

**S14 Fig. 13C-BB spectrum of compound 6.**
(PDF)

**S15 Fig. 13C-DEPT 135 spectrum of compound 6.**
(PDF)

**S16 Fig. 13C-DEPT 90 spectrum of compound 6.**
(PDF)

**S17 Fig. HMBC spectrum of compound 6.**
(PDF)

**S18 Fig. NOESY spectrum of compound 6.**
(PDF)

**S19 Fig. HSQC spectrum of compound 6.**
(PDF)

**S20 Fig. COSY spectrum of compound 6.**
(PDF)

**S21 Fig. EI-MS of compound 8.**
(PDF)

**S22 Fig. HREI-MS of compound 8.**
(PDF)

**S23 Fig. 1H-NMR spectrum of compound 8.**
(PDF)

**S1 Data.**
(CIF)

**S2 Data.**
(CIF)

**S1 File.**
(PDF)

**S2 File.**
(PDF)

**S3 File.**
(PDF)

## Author Contributions

**Conceptualization:** Behram Khan Khoso, Sammer Yousuf, M. Iqbal Choudhary.

**Data curation:** Memoona Bibi, Sammer Yousuf.

**Formal analysis:** Humaira Zafar, Muniza Shaikh.

**Funding acquisition:** M. Iqbal Choudhary.

**Investigation:** Saira Bano, Saba Farooq, Sammer Yousuf.

**Methodology:** Saba Farooq, Behram Khan Khoso.

**Project administration:** M. Iqbal Choudhary.

**Software:** Memoona Bibi, Humaira Zafar, Muniza Shaikh.

**Supervision:** Sammer Yousuf, M. Iqbal Choudhary.

**Writing – original draft:** Saira Bano, Sammer Yousuf.

**Writing – review & editing:** M. Iqbal Choudhary.

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
