## [Decision Letter · Decision Letter 0]

28 Jun 2022

PONE-D-22-16415Anti-leishmanial Physalins - Phytochemcial Investigation, In vitro Evaluation against clinical and MIL Resistant L. tropica strains and In silico Studies PLOS ONE

Dear Dr. Yousuf,

Thank you for submitting your manuscript to PLOS ONE. After careful consideration, we feel that it has merit but does not fully meet PLOS ONE’s publication criteria as it currently stands. Therefore, we invite you to submit a revised version of the manuscript that addresses the points raised during the review process.

We look forward to receiving your revised manuscript.

Kind regards,

Mohammad Shahid, Ph.D.

Academic Editor

PLOS ONE

Journal Requirements:

"“The research leading to these results has, in part, received funding from UK Research and Innovation via the Global Challenges Research Fund under grant agreement ‘A Global Network for Neglected Tropical Diseases’, grant number MR/P027989/1.” 

"“Sammer Yousuf and M. Iqbal Choudhary, researcher leading to these results has, in part, received funding from UK Research and Innovation via the Global Challenges Research Fund under grant agreement ‘A Global Network for Neglected Tropical Diseases’, grant number MR/P027989/1.” 

5. Please ensure that you refer to Figures 5, 7, 8, and 9 in your text as, if accepted, production will need this reference to link the reader to the figure.

6. We note you have included a table to which you do not refer in the text of your manuscript. Please ensure that you refer to Tables 2, 4, and 6in your text; if accepted, production will need this reference to link the reader to the Table.

Reviewers' comments:

Reviewer's Responses to Questions

**Comments to the Author**

1. Is the manuscript technically sound, and do the data support the conclusions?

Reviewer #1: Partly

Reviewer #2: Yes

2. Has the statistical analysis been performed appropriately and rigorously? 

Reviewer #1: No

Reviewer #2: Yes

3. Have the authors made all data underlying the findings in their manuscript fully available?

Reviewer #1: Yes

Reviewer #2: Yes

4. Is the manuscript presented in an intelligible fashion and written in standard English?

Reviewer #1: No

Reviewer #2: Yes

5. Review Comments to the Author

Reviewer #1: General comments

The authors, in continuation of their work, have used different components extracted from physalis minima (isolated and structurally characterized). Anti-leishmanial effects of physalins were explored against L.tropica and L.major, the causative agents of ACL and ZCL, respectively. Although the work is extensive major revisions are needed before being considered for publication.

Specific comments

Title

- Mixed capital and small wording should be corrected according to the journal's style.

Abstract

- 97 countries should be 98 countries (L28-494)

- Exclude 3T3 and BJ in the keywords (L62).

Introduction

- L75, the expected figure for 7 countries is wrong as written (>6,000) are 80% of cases globally?

- Solanaceae (L83) should be written in italic font.

M+M

- Document a standard reference for performing TLC (L102-127).

- Exclude “of the Department of Botany University of Karachi (L131-132). It is a repetition of the above line (L130).

- Present a standard ref. for extraction and isolation processes (134-156).

- The authors did not use an MTT assay to check the viability of promastigotes. Did you check for viability before performing the treatments to make sure they are all viable (motile) because you counted them by eyes?

- RPMT should always be written complete (RPMI-1640, L250)

- The authors visually counted the promastigote stage, why did they use DAPI stain? How did it support the study (L259-260)?

- Correct 37оC (L282).

- Correct 5×104 cells/ml (L283).

- Write the results in the past tense (L423 and other places).

- For cytotoxicity assays on fibroblasts, the authors should calculate the CC50 values, and then to determine the safety index; one should calculate the selectivity index (SI). SI=CC50/IC50≥1, non-toxic.

- If the authors do not use this equation how can one say the compound is safe?

- Be consistent in writing words, the authors have repeatedly used anti-leishmanial in L439 used anti leishmanial (correct).

- Mention why you docked with phosphoglucose isomerase?

- Why the authors did not use a uniform and standard positive control like AmpB throughout the experiments so that one could be able to compare the outcome? All of the positive controls even AmpB, pentamidine, and MIL are the second-line drugs, except AmBisome (a liposomal form of AmpB is used to treat Indian and Mediterranean type kala-azar (visceral leishmaniasis due to L.donovani and L.infantum, respectively). The authors could use meglumine antimoniate (Glucantime®) and if resistance is a common phenomenon then AmpB would be OK, otherwise using different positive controls is not realistic. Miltefosine does not affect L. tropica it is only active at around 83%-85% against L.major and L.infantum.

- Another important thing the authors should know is that: promastigotes are the extracellular stage in the gut of sand flies (biological vectors) and culture media. The authors could use intra-macrophage amastigotes (the clinical stage inside the phagocytic cells in a vertebrate host such as humans and wild mammals). Promastigotes are biochemically and molecularly different from amastigotes (Leishman bodies) in several aspects: they are more resistant and aerobic stage, although extracellular.

- L434, Cytotoxicity of physalin against human fibroblast as given in Table 5 are CC50. These values should be used in the equation above to calculate the SI index as the measure of safety (SI=CC50/IC50≥1, safe), otherwise how can you tell the component is safe?

- This section is not well organized, and it should be titled more specifically according to the journal’s style.

Discussion

- AmpB is not the drug of choice for L.tropica and L. major. It is rather a liposomal form OF AmpB(AmBisome used against VL). It is all right you used other second-choice drugs because you had no access to meglumine antimonite (Glucantime®) or sodium stibogluconate(Pentostam®), but you should have used similar positive controls to be able to compare the activity of each component.

- The authors should not point out the Figs in the discussion (L490).

- Close the sentence in L490.

- I am not export on docking reactions and you pay attention to other referees for this purpose.

- How could you tell that the difference between the test results and the untreated control group is significant, or how could you predict the significant difference among groups. The author should mention the statistical test and define the P-value.

- the discussion is too weak and should be enriched and compared with similar or closely-related issues.

- The docking technique is employed to predict the tentative binding affinity of the ligand-receptor complex ahead of time. So this should preferably be presented at the beginning of the M+M.

Miscellaneous

- The abbreviation should be used at a first appearance for the most commonly used terms such as cutaneous leishmaniasis (CL) in the abstract and other places.

- After using Leishmania tropica or L.major the genus should be used in form of abbreviation throughout the manuscript (L.tropica or L.major). Mixed using of the species wording is used and this should be corrected consistently.

- The Old and New world should be written with capital first initials (i.e., introduction).

- The scientific name of organisms either parasite ( l97-98) or sand flies should be printed in the capital font ( e.i., Lutzomyia ( i.e., L70). In some places, Leishmania is not italic or with a small l (L477, L505).

- In-vitro or in vitro (L478).

- Phlebotomine should be written with small letters in the middle of a sentence (L70).

- The authors initially defined Miltefosine as MIL, then again used Miltefosine in L57. They must be consistent in using abbreviation forms? In some places, miltefosine is written with capital in other places with small (L418).

- Amphotericin B is written with small and capital ”A”(413 and many places). Be consistent in writing words.

Reviewer #2: The manuscript titled: "Anti-leishmanial Physalins - Phytochemcial Investigation, In vitro Evaluation against clinical and  MIL  Resistant L. tropica strains and In silico Studies" was aimed to describe the in vitro cytotoxicity and antileishmanial activity of the physalins from Physalis minima using normal fibroblast (3T3) and BJ (human fibroblast) cells lines and promastigotes L. major and L. tropica. For this, the authors used methodologies widely described and accepted by different authors in the world. The experiments are properly written, which makes them easy to reproduce. The results obtained allow conclusions to be made around the proposed objective and are rightly discussed. In my opinion, this manuscript is sustainable for publication in Plos One.

6. PLOS authors have the option to publish the peer review history of their article (what does this mean?). If published, this will include your full peer review and any attached files.

Reviewer #1: No

Reviewer #2: No

---

## [Decision Letter · Decision Letter 1]

15 Aug 2022

PONE-D-22-16415R1Anti-leishmanial Physalins - Phytochemcial Investigation, In vitro Evaluation against clinical and MIL Resistant L. tropica strains and In silico StudiesPLOS ONE

Dear Dr. Yousuf,

Thank you for submitting your manuscript to PLOS ONE. After careful consideration, we feel that it has merit but does not fully meet PLOS ONE’s publication criteria as it currently stands. Therefore, we invite you to submit a revised version of the manuscript that addresses the points raised during the review process.

We look forward to receiving your revised manuscript.

Kind regards,

Mohammad Shahid, Ph.D.

Academic Editor

PLOS ONE

Journal Requirements:

Reviewers' comments:

Reviewer's Responses to Questions

**Comments to the Author**

1. If the authors have adequately addressed your comments raised in a previous round of review and you feel that this manuscript is now acceptable for publication, you may indicate that here to bypass the “Comments to the Author” section, enter your conflict of interest statement in the “Confidential to Editor” section, and submit your "Accept" recommendation.

Reviewer #1: (No Response)

2. Is the manuscript technically sound, and do the data support the conclusions?

Reviewer #1: Yes

3. Has the statistical analysis been performed appropriately and rigorously? 

Reviewer #1: Yes

4. Have the authors made all data underlying the findings in their manuscript fully available?

Reviewer #1: Yes

5. Is the manuscript presented in an intelligible fashion and written in standard English?

Reviewer #1: No

6. Review Comments to the Author

Reviewer #1: Specific comments

-The title still needs corrections as I displayed by track changes.

-In the abstract, the authors abbreviated cutaneous leishmaniasis in two places (L 27-30).

- The authors did not use an MTT assay to check the viability of promastigotes. Did they check for viability before performing the treatments to make sure they are all viable (motile) because they counted them by eyes?

- The authors visually counted the promastigote stage, why did they use DAPI stain? How did it support the study?

- Mention why you docked with phosphoglucose isomerase.

- Why the authors did not use a uniform and standard positive control like AmpB throughout the experiments so that one could be able to compare the outcome? All of the positive controls even AmpB, pentamidine, and MIL are the second-line drugs, except AmBisome (a liposomal form of AmpB is used to treat Indian and Mediterranean type kala-azar (visceral leishmaniasis due to L.donovani and L.infantum, respectively). The authors could use sodium stibogluconate (Pentostam®) or meglumine antimoniate (Glucantime®) and if resistance is a common phenomenon then AmpB would be OK, otherwise using different positive controls is not realistic. Miltefosine does not affect L. tropica it is only active at around 83%-85% against L.major and L.infantum.

- AmpB is not the drug of choice for L.tropica and L. major. It is rather a liposomal form OF AmpB (AmBisome used against VL). It is all right you used other second-choice drugs because you had no access to meglumine antimonite (Glucantime®) or sodium stibogluconate(Pentostam®), but you should have used similar positive controls to be able to compare the activity of each component.

- I am not export on docking reactions and you pay attention to other referees for this purpose.

- How could you tell that the difference between the test results and the untreated control group is significant, or how could you predict the significant difference among groups? The author should mention the statistical test and define the P-value. In the results whenever a significant level is seen the p-value should be pointed out.

- The docking technique is employed to predict the tentative binding affinity of the ligand-receptor complex ahead of time. So this should preferably be presented at the beginning of the M+M.

- After using Leishmania tropica or L.major the genus should be used in form of abbreviation throughout the manuscript (L.tropica or L.major). Mixed using of the species wording is used and this should be corrected consistently.

- The Old and New World should be written with capital first initials (like World).

- The scientific name of organisms either parasites or sand flies should be printed in the capital font ( e.i., Lutzomyia). In some places, Leishmania is not italic or with a small l.

- Subtitles are not uniformly written and they are written with mixed capital and small ones, whereas the journal style is different. I advise the authors once again to review the journal’s instructions or one of the newly published Plos One articles and follow through and make the corrections.

7. PLOS authors have the option to publish the peer review history of their article (what does this mean?). If published, this will include your full peer review and any attached files.

Reviewer #1: No

---

## [Author Response · Author response to Decision Letter 1]

25 Aug 2022

The response are uploaded as word document.

---

## [Editor Report · Decision Letter 2]

30 Aug 2022

Anti-leishmanial physalins - phytochemcial investigation, in vitro evaluation against clinical and MIL- Resistant L. tropica strains and in silico studies

PONE-D-22-16415R2

Dear Dr. Yousuf,

We’re pleased to inform you that your manuscript has been judged scientifically suitable for publication and will be formally accepted for publication once it meets all outstanding technical requirements.

Kind regards,

Mohammad Shahid, Ph.D.

Academic Editor

PLOS ONE
---

## [Editor Report · Acceptance letter]

5 Sep 2022

PONE-D-22-16415R2 

Anti-leishmanial physalins - phytochemical investigation, *in vitro* evaluation against clinical and MIL-resistant L. *tropica* strains and *in silico* studies 

Dear Dr. Yousuf:

I'm pleased to inform you that your manuscript has been deemed suitable for publication in PLOS ONE. Congratulations! Your manuscript is now with our production department. 

Kind regards, 

on behalf of

Dr. Mohammad Shahid 

Academic Editor

PLOS ONE